# Plant-Wide Modeling and Economic Analysis of Monoethylene Glycol Production

Md Emdadul Haque [1], Namit Tripathi [2], Srinivas Palanki [1,*], Qiang Xu [3] and Krishna D. P. Nigam [4]

1 Department of Chemical & Biomedical Engineering, West Virginia University, Morgantown, WV 26506, USA
2 Linde Inc., 1585 Sawdust Rd., The Woodlands, TX 77380, USA
3 Dan F. Smith Department of Chemical Engineering, Lamar University, Beaumont, TX 77710, USA
4 Indian Institute of Technology, Delhi Hauz Khas, New Delhi 110016, India
* Correspondence: srinivas.palanki@mail.wvu.edu; Tel.: +1-304-293-9364

**Abstract:** Monoethylene glycol (MEG) is used to produce polyester fibers and polyethylene terephthalate resins. It is also utilized in antifreeze, pharmaceuticals, and cosmetics applications. In this research, we consider the development of a novel process plant that produces MEG from ethylene. The proposed ethylene-to-ethylene oxide (EO) plant is integrated with an EO-to-MEG plant to reduce utility costs and recover high-value products. Energy-saving opportunities are analyzed via heat integration tools. Furthermore, a multitube glycol reactor is used in conjunction with a novel MTO catalyst in the ethylene-to-EO reactor. Our results demonstrate that the integrated EO/EG plant produces ethylene glycols with that same purity and product recovery as conventional designs. A comparative economic assessment based on a 200,000 t/y plant indicates that process integration techniques can reduce costs significantly.

**Keywords:** process modeling; process integration; heat integration; economic analysis; ethylene glycol





## 1. Introduction

Monoethylene glycol (MEG) is used to produce polyester fibers and polyethylene terephthalate resins. It is also utilized in antifreeze, pharmaceuticals, and cosmetics applications [1]. The global MEG demand was estimated to be 26.9 million tons in 2016. This market is predicted to grow at a rate of 6.0% to 6.5% annually between 2017 and 2024 [2]. The conventional route for the manufacture of MEG first involves oxidizing ethylene to ethylene oxide (EO) and then hydrolyzing EO to MEG [1]. Traditionally, the conversion of ethylene to EO and the conversion of EO to MEG is achieved in separate process plants. However, recently, there has been a move to integrate EO/EG plants so that almost all the EO produced is converted to glycols [3]. Plant integration can result in significant savings in utilities. Furthermore, it is possible to recover all bleed streams as high-grade products instead of lower-grade products, as in the case of non-integrated plants [3]. There are also opportunities for integrated plants to reduce capital expenditure (CAPEX), operating expenditure (OPEX), and environmental impact.

EO is a very reactive chemical because opening of its ring shape releases significant amounts of energy [2]. Its explosion range varies from 3.6% to 100% of EO in air [4]. The conventional method of producing EO involves an epoxidation reaction whereby ethylene is epoxidized to EO using oxygen as an oxidant. The reaction is catalyzed by a silver-based catalyst at a temperature of 200 °C to 260 °C and at 1 MPa to 3 MPa [5]. A catalytic reaction of EO with water at higher temperature results in the formation of the main product, MEG, and the byproducts diethylene glycol (DEG) and triethylene glycol (TEG). Because the reaction occurs at high temperatures and oxygen is present in the reactor, it is possible for ethylene and EO to react, resulting in the formation of unwanted water and carbon dioxide. This undesirable reaction results in an estimated yearly loss of as much as USD 1.2 billon [6,7]. Additionally, because ethylene and EO reactions are

exothermic in nature, the risk of a runaway reaction is very high [6]. This process also generates approximately 3.4 million tons of carbon dioxide per year, which contributes to greenhouse gas emissions [7,8]. Issues with EO handling can result in reduced purity and off-spec production of MEG in EO/EG integrated plants. These factors provide the motivation to seek process improvement opportunities with respect to the production of MEG from ethylene.

A conventional approach of industrial production of EO from ethylene is epoxidation reaction, as explained by Rebsdat and Mayer (2012) [2]. A useful summary of this ethylene oxidation reaction system is provided by Nawaz et al. (2016) [9], and most studies have used Langmuir–Hinshelwood–Hougen–Watson (LHHW)-type kinetics. The LHHW approach assumes that all active sites are energetically uniform, and upon adsorption, adsorbed species do not interact with already adsorbed species. Active sites have similar kinetic and thermodynamic characteristics, and the entropy and enthalpy of adsorption are constant and not functions of the adsorbed amount. Species adsorption restricts itself to only monolayer coverage, and the rate of adsorption is proportional to the concentration of the active sites not occupied (empty) and the partial pressure of the component in the gas phase [9,10]. In 1990s, the Westerterp group published a series of papers wherein extensive experiments and kinetic model development for EO were presented. Their study focused on kinetics of ethylene in the presence of excess air on an unpromoted silver catalyst supported on alumina in the absence of chlorinated hydrocarbon moderators [10–13]. In 2009, it was discovered that the epoxidation reaction of ethylene to EO could be achieved in a gas-expanded liquid phase using methyltrioxorhenium (MTO) as the homogenous catalyst, pyridine-N-oxide as the catalyst promoter, hydrogen peroxide ($H_2O_2$) as the oxidant, and methanol as the solvent [14,15]. The reaction is carried in a continuous stirred reactor (CSTR) close to the critical point of ethylene (Pc = 5.042 MPa; Tc = 9.2 °C) [9,10]. Under these conditions, ethylene dissolves better in the solvent and reacts with $H_2O_2$ at the surface sites of the MTO catalyst without producing any $CO_2$. The application of an MTO catalyst in EO production has been studied recently in the literature [16–19]. In this modified scheme, the excess ethylene left over after the epoxidation reaction is recycled back to the main epoxidation reactor. The excess $H_2O_2$ is decomposed to $H_2O$ and $O_2$ in a separate unit, resulting in a safer operation.

Whereas studies have been conducted on the use of an MTO catalyst to convert ethylene to EO, a survey of the literature indicates that no studies have been conducted to analyze integrated plants utilizing an MTO catalyst to produce MEG from ethylene. Furthermore, opportunities for process integration in such plants have not been explored beyond the new catalyst development described in the previous paragraph. There are opportunities to consider novel reactor configurations that could result in dramatic improvements to process yields [20,21].

In this research, a novel plant to produce MEG from ethylene is designed using process integration techniques. The proposed plant represents two design innovations: (i) the ethylene-to-EO plant is integrated with an EO-to-MEG plant to reduce utility costs and recover high-value products and (ii) a multitube glycol reactor is used in conjunction with a novel MTO catalyst in the ethylene-to-EO reactor for the purpose of energy savings and cost reduction. This integrated system has the potential to produce MEG with a purity of 99.9% with virtually no accumulation of highly-flammable EO in the process. Furthermore, the production of carbon dioxide is eliminated.

## 2. Process Description and Simulation

### 2.1. Simulation Packages

Aspen Plus V10 [22] and its economic and energy analyzer packages were utilized in this research. In this study, two thermodynamic packages were used to develop the proposed model. In particular, the non-random two-liquid model (NRTL) was used for the EO process, as suggested in the literature, and the cubic plus association equation of state (CPA EOS) was used for the MEG process [16,23,24]. The interaction parameters to predict

the vapor liquid equilibrium of ethylene, methanol, ethylene oxide, and methanol were adopted from Lee et al. [16].

### 2.2. Production Process

The process to convert ethylene to EO consists of two reactors (REAC-1 and REAC-2), a flash drum (V-1), and three distillation columns (C-1, C-2, and C-3), as shown in Figure 1. The feed is introduced to the first reactor (REAC-1). Ethylene and hydrogen peroxide react in the presence of the MTO catalyst to produce ethylene oxide and water in the reactor. The feed specifications used in this process are listed in Table 1. The reaction parameters are taken from the Ghanta et al. [17] and are listed Table 2 as reaction 1. The residence time of the reactor is assumed to be 0.25 h, as recommended by Ghanta et al. [17]. The vapor outlet of the flash drum (V-1), which is mostly ethylene, is compressed and recycled back to the first reactor after cooling down to the feed temperature. The liquid outlet stream of the flash drum (V-1) enters the second reactor (REAC-2), and the remaining $H_2O_2$ is decomposed to water and oxygen before entering the distillation columns. The reaction parameters are taken from Schumb et al. [25] and are listed in Table 2 as reaction 2.

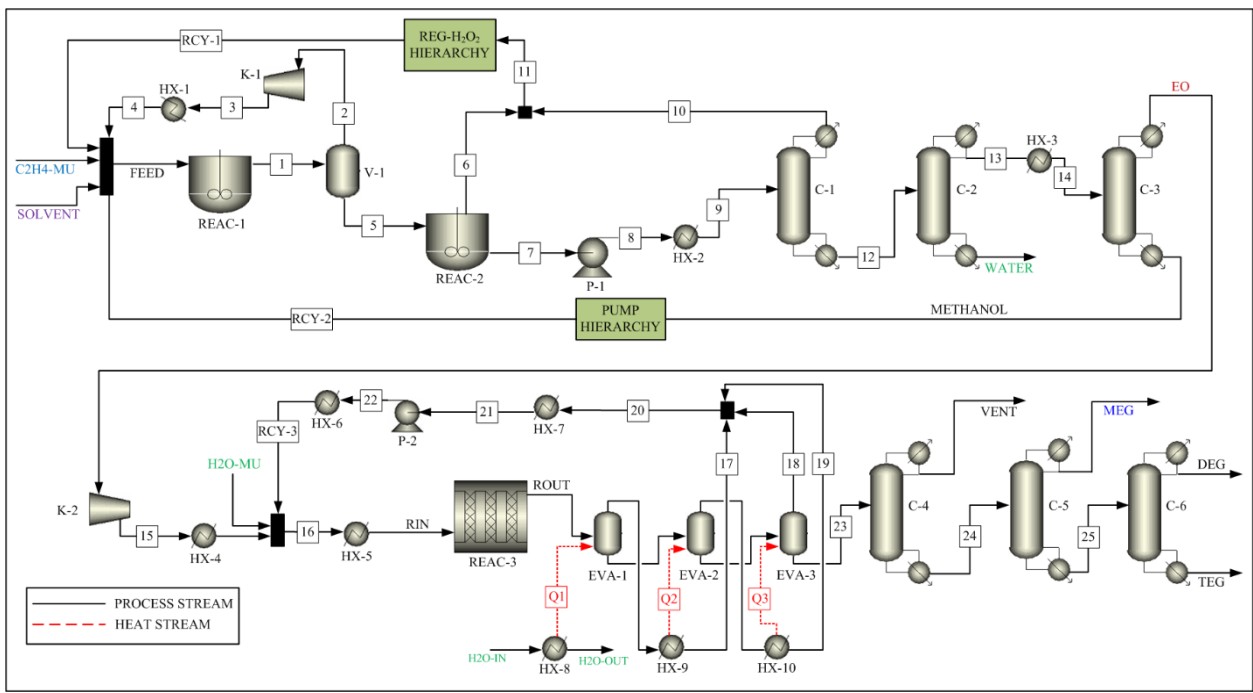

**Figure 1.** Process Flow Sheet for the Integrated Process.

**Table 1.** Feed specifications of the developed process streams.

| Parameter | Feed Stream |
|---|---|
| Pressure (MPa) | 5 |
| Temperature (°C) | 40 |
| Flow Rate (kg/h) | 160,263.0 |
| *Composition (mass%)* | |
| Hydrogen peroxide | 12.21 |
| Methanol | 64.21 |
| Ethylene | 10.97 |
| Ethylene oxide | 0.40 |
| Oxygen | 0 |
| Water | 12.21 |
| **Total** | **100** |

**Table 2.** Reaction parameters [17,25].

| Reaction | Reaction Type | Reaction Rate | Rate Constant | Reference Temp. | Activation Energy |
|---|---|---|---|---|---|
| 1 | LHHW | $-r = k \, C_{H_2O_2} \, C_{EL} \, C_{cat.}$ | $6.2 \times 10^{-6} \quad s^{-1}$ | 20 °C | $57 \, \frac{kJ}{mol}$ |
| 2 | Power law | $-r = k \, C_{H2O2}^{2}$ | $62.5 \quad \frac{m^3}{kmol \cdot s}$ | 140 °C | $52.7 \, \frac{kJ}{mol}$ |

A train of three distillation columns in series is used to separate the product and recycle the unreacted reactant and solvent. The first column (C-1) is a packed column with a partial condenser, in which ethylene oxide is recovered in the bottom stream. The second column (C-2) is a packed column with a total condenser, in which water is recovered in the bottom stream, whereas the remaining ethylene oxide and methanol are recovered as top products. The third column (C-3) is a packed column with a partial condenser, which is used to recover ethylene oxide in the top stream. The methanol solvent is recovered in the bottom stream and is recycled back to the first reactor.

A series of four pumps and four heaters is used to raise the stream pressure and temperature to the required feed conditions in the solvent recycle stream. These are represented in the PUMP sub-flowsheet block. A conversion reactor is utilized to convert the generated oxygen to hydrogen peroxide, which is recycled back to the first reactor. A heater block is used to raise the stream pressure and temperature to operation conditions. This is represented by the REG-$H_2O_2$ sub-flowsheet block.

As shown in Figure 1, the process to convert EO to MEG, DEG, and TEG consists of a reactor (REAC-3), a triple-effect evaporator (EVA-1, EVA-2, EVA-3), and three distillation columns (C-4, C-5, C-6). As the conversion of EO to glycols take place in the liquid phase, it is common practice to use a single pipe reactor with a very long pipe length or multiple reactors in series [26]. However, in this simulation, a multitube reactor (REAC-3) is used as a compact reactor, which will replace a long pipe or multiple reactors. The use of a multitube reactor reduces the plant footprint and provides an opportunity to reduce the fixed capital investment cost, which in turn leads to the possibility of reduced product cost while maintaining the process profit margin. Multitube reactor systems are usually used for exothermic reactions. Such systems maintain a uniform temperature profile across the catalyst bed length [27,28], which improves not only the process performance [29] but also the catalyst performance and catalyst life. The reaction is highly exothermic, resulting in a significant increase in the reactor outlet stream temperature. This outlet stream is sent to a triple-effects evaporator for purification and is modeled as a series of three flash unit blocks with heat supplies Q1, Q2, and Q3. The glycols and water with impurities are transferred to a dehydrator for further processing. The overhead vapor streams of the triple-effect evaporator are collected and condensed as a recycle stream. The bottom liquid stream of the triple-effect evaporator (stream #23) is transferred to the dehydrator (C-4) for removal of waste stream, which consists of water and aldehydes. The product stream from the dehydrator is sent to the MEG distillation column (C-5). The column is operated in a vacuum to distilling high-grade pure polyester MEG of 99.99% purity. Other heavier glycols are transferred to a second column (C-6) for further processing of DEGs and TEGs.

*2.3. Heat Exchange Network*

A heat exchange network (HEN) is utilized for the ethylene glycol process to minimize the cost of both hot and cold utilities. The process streams involved in the HEN are tabulated in Table 3. The objective function minimizes the total utility costs, and the minimum temperature ($\Delta T_m$) is assumed to be 3.0 °C.

**Table 3.** Process streams involved in HEN.

| Stream ID | Block | Stream Type | $T_{in}$ (°C) | $T_{out}$ (°C) | Duty (MW) |
|---|---|---|---|---|---|
| 1 | HX-2 | Cold Stream | 40.3 | 137.8 | 23.66 |
| 2 | Reboiler@C-1 | Cold Stream | 135.3 | 138.7 | 7.88 |
| 3 | Reboiler@C-2 | Cold Stream | 162.0 | 162.1 | 143.05 |
| 4 | Reboiler@C-3 | Cold Stream | 64.0 | 64.5 | 117.58 |
| 5 | HX-5 | Cold Stream | 19.5 | 171.1 | 59.08 |
| 6 | Reboiler@C-4 | Cold Stream | 159.1 | 165.8 | 94.25 |
| 7 | Reboiler@C-5 | Cold Stream | 200.2 | 202.3 | 10.65 |
| 8 | Reboiler@C-6 | Cold Stream | 192.1 | 193.6 | 0.23 |
| 9 | REG-$H_2O_2$.R-103 | Cold Stream | 74.8 | 75.3 | 1.54 |
| | ***Total Hot Utility Required*** | | | | ***457.93*** |
| 10 | HX-1 | Hot Stream | 205.7 | 40.0 | $-7.284 \times 10^5$ |
| 11 | REAC-1 | Hot Stream | 41.1 | 40.0 | $-1.145 \times 10^8$ |
| 12 | REAC-2 | Hot Stream | 40.0 | 39.5 | $-5.349 \times 10^6$ |
| 13 | REG-HX | Hot Stream | 74.8 | 40.0 | $-5.385 \times 10^5$ |
| 14 | Condenser@C-1 | Hot Stream | 133.2 | 48.8 | $-2.889 \times 10^7$ |
| 15 | Condenser@C-2 | Hot Stream | 119.5 | 119.3 | $-4.982 \times 10^8$ |
| 16 | HX-3 | Hot Stream | 119.3 | 62.3 | $-2.545 \times 10^7$ |
| 17 | Condenser@C-3 | Hot Stream | 46 | 20.3 | $-3.881 \times 10^8$ |
| 18 | HX-4 | Hot Stream | 276.1 | 19.4 | $-1.964 \times 10^7$ |
| 19 | Condenser@C-4 | Hot Stream | 53.4 | 41.3 | $-3.833 \times 10^8$ |
| 20 | Condenser@C-5 | Hot Stream | 138.7 | 136.8 | $-3.873 \times 10^7$ |
| 21 | Condenser@C-6 | Hot Stream | 133.7 | 123.2 | $-9.392 \times 10^5$ |
| 22 | HX-7 | Hot Stream | 98.0 | 19.4 | $-2.464 \times 10^8$ |
| 23 | HX-6 | Hot Stream | 96.2 | 19.4 | $-3.566 \times 10^7$ |
| 24 | Pump-HX-1 | Hot Stream | 64.1 | 40.0 | $-7.674 \times 10^6$ |
| 25 | Pump-HX-2 | Hot Stream | 40.4 | 40.0 | $-1.092 \times 10^5$ |
| 26 | Pump-HX-3 | Hot Stream | 41.1 | 40.0 | $-3.278 \times 10^5$ |
| 27 | Pumps-HX-4 | Hot Stream | 41.4 | 40.0 | $-4.098 \times 10^5$ |
| | ***Total Cold Utility Required*** | | | | ***526.05*** |

The HEN design utilizes a pinch analysis technique to compute overall matches with heat load, surface area, and cost target in the optimization algorithm. The δ function, which is used in the fitness calculation, is expressed as [30]:

$$\delta_{ijk} = \left| 1 - \left| \frac{q_{ijk}}{Q_k} \frac{\frac{1}{CP_i} - \frac{1}{CP_j}}{\frac{1}{CP_{(Hot-quasi)k}} - \frac{1}{CP_{(Cold-quasi)k}}} \right| \right|; \text{ when } CP_{(Hot-quasi)k} \neq CP_{(Cold-quasi)k} \quad (1)$$

and

$$\delta_{ijk} = \frac{q_{ijk}}{Q_k} \frac{CP_{(Hot-quasi)k}}{CP_i} \left| 1 - \frac{CP_i}{CP_j} \right|; \text{ when } CP_{(Hot-quasi)k} = CP_{(Cold-quasi)k} \quad (2)$$

Where:
$CP_i$ = the heat capacity flowrate of hot stream $i$;
$CP_j$ = the heat capacity flowrate of cold stream $j$;
$q_{ijk}$ = the heat load between hot stream $i$ and cold stream $j$ in the kth block;
$Q_k$ = enthalpy change of the kth block;
$CP_{(Hot-quasi)k}$ = the heat capacity flow rate of the hot quasi-composites in the kth block; and
$CP_{(Cold-quasi)k}$ = the heat capacity flow rate of the cold quasi-composites in the kth block.

The optimal HEN design is determined by minimizing an objective function, which is defined as:

$$min \sum_{ij} \delta_{ij} \qquad (3)$$

This minimization problem is formulated as a mixed-integer linear program (MILP). For this MILP, it is assumed that there is an equal number (*n*) of hot and cold streams. Furthermore, $\delta_{ij}$ is defined as the fitness to the quasi-composites according to a match between hot stream *i* and cold stream *j*. Binary variables ($x_{ij}$) are used to indicate the existence of a match between hot stream *i* and cold stream *j*; therefore, $x_{ij} = 0$ implies that there is no match between the hot and cold streams, and a utility stream is required. Thus, Equation (3) can be converted into the following MILP problem, as shown in [30]:

$$
\begin{aligned}
&min \sum_{i,j=1}^{n} \delta_{ij}.x_{ij} \\
&\sum_{i=1}^{n} x_{ij} = 1 \quad j = 1, \ldots, n \\
&\sum_{j=1}^{n} x_{ij} = 1 \quad i = 1, \ldots, n \\
&x_{ij} = 0, 1 \quad i, j = 1, \ldots, n
\end{aligned}
\qquad (4)
$$

In this method, both the overall surface area and the number of units are minimized. This results in the most suitable set of matches rather than a single match. Because the MILP model assumes an equal number of hot and cold streams, it is necessary to use dummy elements with zero assignment to apply this method to cases in which the number of hot and cold streams is not equal.

Aspen Energy Analyzer V11.0 (AEA) [31] was used to develop and optimize the HEN. Three cases were considered. The base case considers the overall plant that converts ethylene to EO and EO to EG with no heat integration. The second case considers a scenario in which the overall plant is divided into two sub-plants—(1) ethylene to EO and (2) EO to EG—and heat integration is achieved only within the two sub-plants, but no heat integration occurs between the two sub-plants. The third case considers a scenario where in which integration is performed for the entire flow sheet.

*2.4. Equipment Sizing*

The sizes of flash drums, heat exchangers, and pumps are defined by Aspen Process Economic Analyzer (APEA) [31]. Prices of stream and process utilities [32,33] are required by APEA. Once the APEA tab is turned on, most of the equipment data are assumed based on the process parameters of the simulation. Detailed equipment sizing parameters are provided in Tables 4–7. Column specifications (e.g., reflux ratio, number of stages, etc.) were decided after running several simulations with different specifications, and the parameter set with the best results (Table 4) was adopted as the final set of specifications. The multitube reactor parameters shown in Table 5 were decided based on literature values [34]. The number of tubes and tube length were decided so that total tube length would be the same as that of a conventional single-tube reactor. Sizing parameters for the CSTR reactor and evaporators were set by ASPEN through APEA, as shown in Tables 6 and 7, respectively. A concentration of $H_2O_2$ in the 8% to 28% range is rated as a class 1 oxidizer and an unstable substance [32]. Therefore, the selection of construction materials for the reactors (REAC-1, REAC-2, and REG-$H_2O_2$) should be carefully considered and selected as 304 stainless steel (304 SS). This material is suitable for temperatures up to 49 °C, whereas the reactor temperature is below this value (40 °C). All other construction materials are selected as carbon steel (CS).

**Table 4.** Column sizing and specification parameters.

| Columns | | | | | | |
|---|---|---|---|---|---|---|
| **Name** | **C-1** | **C-2** | **C-3** | **C-4** | **C-5** | **C-6** |
| Mapping | C1 | C2 | C3 | 35 | C5 | C6 |
| Materials of construction | CS | CS | CS | CS | CS | CS |
| Number of stages | 4 | 35 | 100 | 35 | 24 | 29 |
| Reflux ratio | 2 | 10 | 10 | 0.055 | 0.4 | 0.985 |
| Feed tray | 4 | 28 | 7 | 2 | 13 | 18 |
| Tray spacing (m) | 0.76 | 0.61 | 0.61 | 0.61 | 0.61 | 0.61 |
| Tray type | Sieve | Sieve | Sieve | Sieve | Sieve | Sieve |
| Column diameter (m) | 1.52 | 3.96 | 4.88 | 3.96 | 1.22 | 5.49 |

**Table 5.** Equipment sizing parameters for multitube reactors.

| Name | REAC-3 |
|---|---|
| No. of pipes | 20 |
| Pipe length (m) | 15.24 |
| Pipe diameter (m) | 0.30 |
| Inlet pressure (MPa) | 3.5 |
| Inlet temperature (°C) | 171.1 |
| Outlet temperature (°C) | 204.4 |

**Table 6.** Equipment sizing parameters for CSTR reactors.

| Name | REAC-1 | REAC-2 | REG-$H_2O_2$ |
|---|---|---|---|
| Materials of construction | 304 SS | 304 SS | 304 SS |
| Liquid volume ($m^3$) | 58.55 | 1.53 | 0.84 |
| Vessel diameter (m) | 2.74 | 0.76 | 0.61 |
| Vessel tangent-to-tangent height (m) | 9.91 | 3.35 | 2.90 |
| Design gauge pressure (MPa) | 5.24 | 0.57 | 0.57 |
| Design temperature (°C) | 121.1 | 121.1 | 121.1 |

**Table 7.** Equipment sizing parameters for the evaporator.

| Vertical Vessel | | |
|---|---|---|
| **Name** | **EVA-1** | **EVA-2** | **EVA-3** |
| Materials of construction | CS | CS | CS |
| Liquid volume ($m^3$) | 27.52 | 18.32 | 11.21 |
| Vessel diameter (m) | 2.29 | 1.98 | 1.83 |
| Vessel tangent-to-tangent height (m) | 6.71 | 5.94 | 4.27 |
| Design gauge pressure (MPa) | 3.74 | 1.47 | 0.97 |
| Design temperature (°C) | 264.5 | 226.3 | 208.1 |
| Operating temperature (°C) | 236.7 | 198.6 | 180.3 |

*2.5. Capital and Production Costs*

In addition to the fixed capital investment, the raw material and utility costs are also considered for the production cost [33]. Chemical prices per pound are obtained from the Chemical Market Report and are listed in Table 8 [34–36]. The utilities used in the process design and overall process economic evaluation are shown in Table 9.

**Table 8.** Raw material and product prices.

| Classification | Material Name | Unit Cost (USD/kg) |
|---|---|---|
| Raw Material | Ethylene | 0.71 |
| | Ethylene oxide | 1.74 |
| | Hydrogen peroxide | 0.82 |
| | Methanol | 2.07 |
| | Methyltrioxorhenium (MTO)[5] | 11,022.93 |
| Product | MEG | 1.57 |
| | DEG | 0.86 |
| | TEG | 1.54 |

**Table 9.** Overall economic evaluation.

| Class | Description | Cost (USD) |
|---|---|---|
| CAPEX | Reactors | 2,984,100 |
| | Columns | 77,745,900 |
| | Compressors | 7,388,700 |
| | Heat exchangers | 4,352,800 |
| | Pumps | 910,100 |
| | Vessels and tanks | 1,378,600 |
| | Auxiliaries | 47,871,600 |
| | Total fixed capital investment | 142,631,800 |
| OPEX | Waste treatment (USD/year) | 9,402,844 |
| | Operating cost (USD/year) | 54,228,240 |
| Revenue | Revenue from sales (USD/year) | 216,352,152 |

## 3. Simulation Results and Discussion

Ethylene gas in the feed stream mixes with the methanol solvent and dissolves in the liquid phase, where the reaction takes place in first reactor. The oxidant $H_2O_2$, which is a 50% $H_2O_2/H_2O$ feed composition, is stable and does not decompose to water and oxygen under reactor conditions. Thus, the operation of the first reactor is inherently safer than the operation of the conventional ethylene epoxidation in the gas phase at the high temperatures and pressures. Furthermore, the dissolved ethylene and the oxidant $H_2O_2$ react at the catalyst site. In the presence of high concentrations of $H_2O_2$, the MTO catalyst forms a di(peroxo)rhenium complex intermediate, which reacts with ethylene to produce EO. However, because the activity of the intermediate complex decreases with an increase in water concentration in the mixture, ethylene is fed in excess quantity, and $H_2O_2$ is a reaction-limiting reactant. Under these conditions, only the epoxidation reaction proceeds, and the EO reaction selectivity exceeds 99% [17].

Next, the depressurization process in the flash drum is conducted without the decomposition of unreacted hydrogen peroxide into water and oxygen. This is done to prevent any fire or explosion scenarios. A serious hazardous situation can result if the unreacted $H_2O_2$ is allowed to decompose in the presence of EO and ethylene in the reboilers of the distillation train after the reactors. Therefore, the unreacted hydrogen peroxide is decomposed in the REAC-2 reactor before being sent to the separation stages.

The EO stream is sent to the ethylene glycol production section as a feed stream. This stream mixes with water, and the mixture travels to the multitube reactor, where an exothermic reaction occurs. A variety of design parameters were tested for the multitube reactor configuration. The objective was to determine a set of parameters that results in the utilization of as much EO as possible. The final multitube configuration is shown in Table 6. The reactor tube length can be validated in two ways: first, the optimum length is the length beyond which there is a negligible increase in temperature. Because the glycol formation reaction is an exothermic reaction, a zero rate of change in temperature indicates

that the reaction is complete, as shown in Figure 2a. The reactor tube length can also be validated by observing the progression of required product concentration along the tube length. If the product concentration is unchanged after a certain length, then the point at which the concentration stops increasing is the optimum length for the reactor tube, as shown in Figure 2b. Figure 2b shows that 97% of EO is utilized.

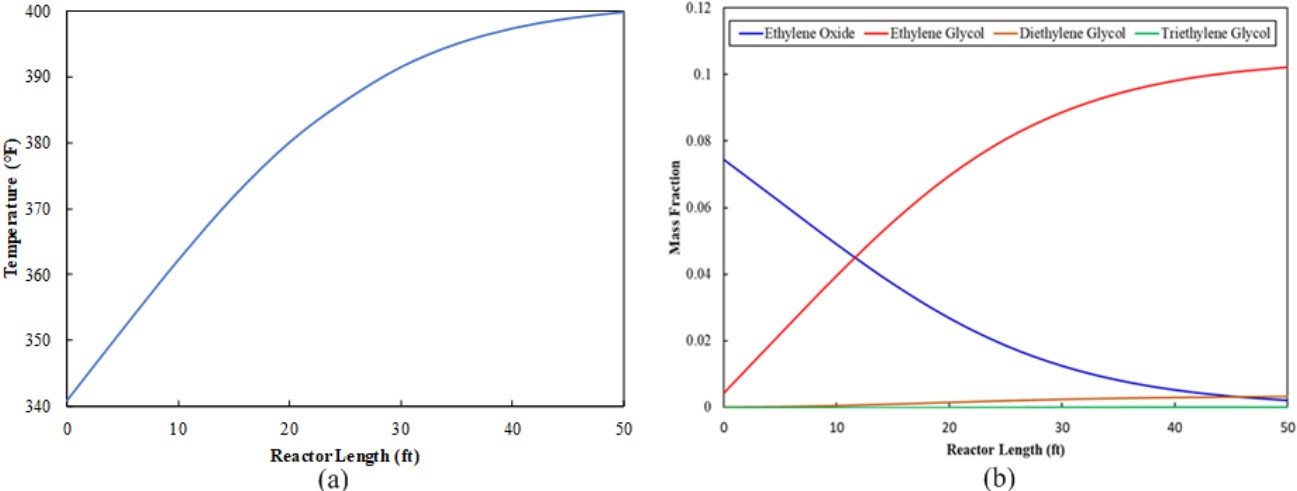

**Figure 2.** Multitube reactor profile along the tube: (**a**) temperature progression and (**b**) composition distribution.

Sizing parameters for the multitube reactor are shown in Table 5. The multitube reactor configuration utilizes a small footprint. Another advantage of the multitube configuration for exothermic reactions is the fact that it is easy to run the reactor under isothermal conditions. Table 10 shows a comparison of the outlet composition at different temperatures in the range of 171.1 °C to 215.6 °C at the outlet of the multitube reactor (REAC-3) if it runs isothermally. The MEG concentration reaches its maximum at an isothermal temperature of 171.1 °C. This strengthens the possibility of incorporating a multitube reactor as a part of process integration to modify the existing conventional EO hydrolysis process.

**Table 10.** Composition comparison for a multitube reactor (REAC-3) under isothermal conditions.

| Composition (wt%) | 171.1 °C | 182.2 °C | 193.3 °C | 204.4 °C | 215.6 °C |
|:---:|:---:|:---:|:---:|:---:|:---:|
| MEG | 11.13 | 11.07 | 10.67 | 10.23 | 9.70 |
| DEG | 0.379 | 0.375 | 0.357 | 0.335 | 0.310 |
| TEG | 0.007 | 0.007 | 0.007 | 0.006 | 0.006 |

Figure 3 shows the ethylene glycol composition profile along the stages of column C-5. The mass fraction of MEG at the top of the column is 99.9%, whereas it is negligible at the bottom, which indicates appropriate separation of MEG from impurities and validates the operating and sizing parameters for column C-5. Similarly, Figure 4 shows that the mass fraction of DEG at the top of column C-6 is 99.9%, whereas it is negligible at the bottom, which validates the operation and sizing parameters for column C-6. Table 11 shows the stream results of the developed integrated process.

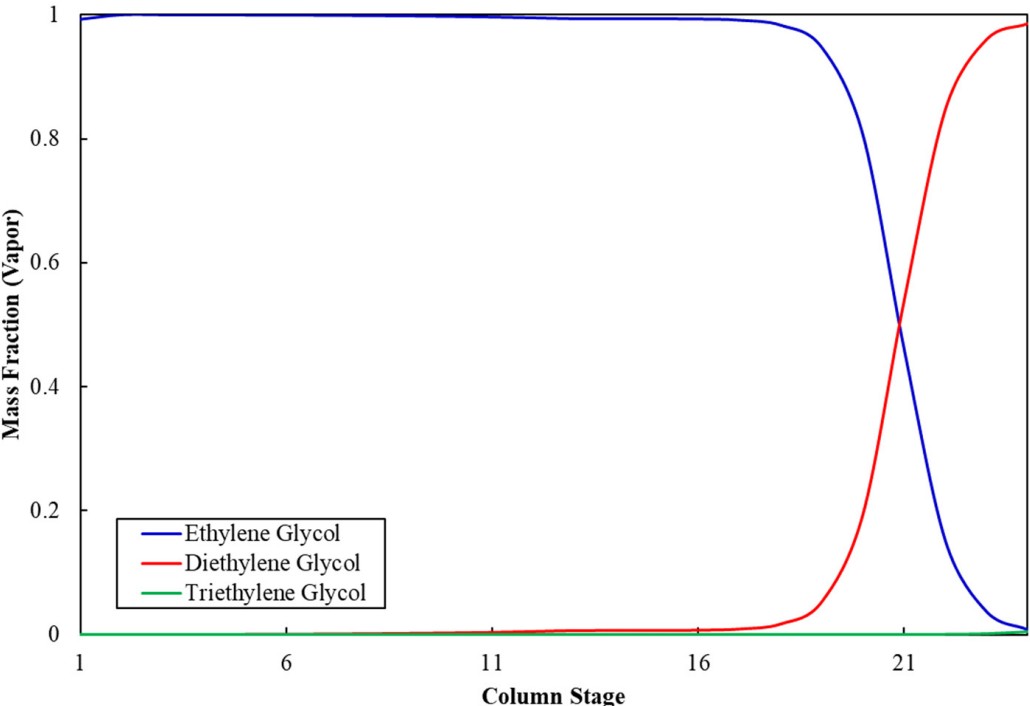

**Figure 3.** Ethylene glycol composition profile in column C-5.

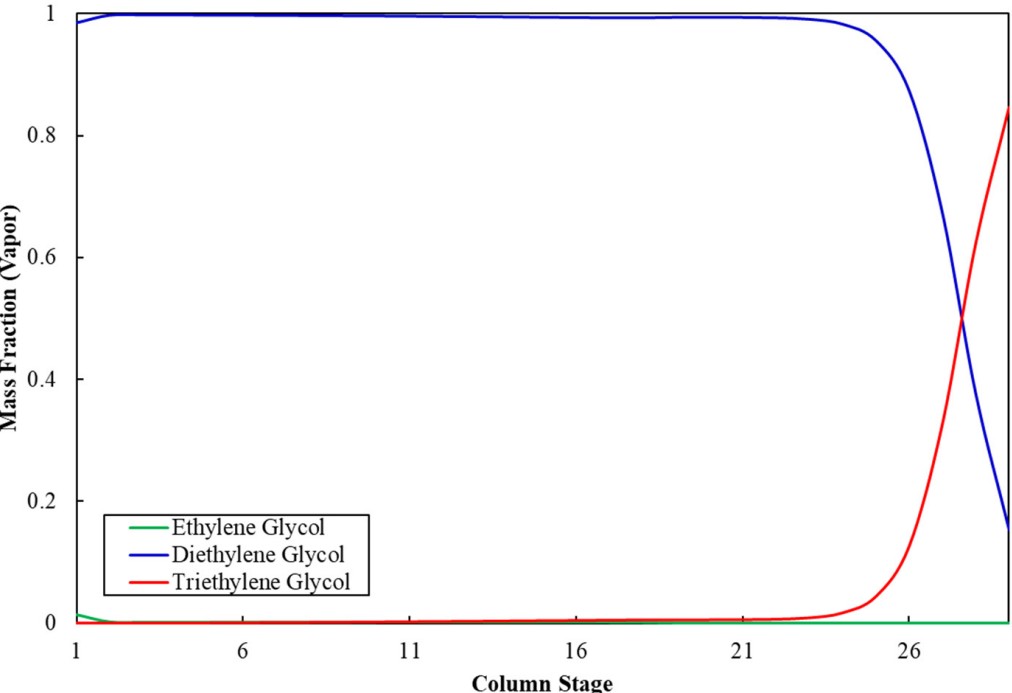

**Figure 4.** Diethylene glycol composition profile in column C-6.

**Table 11.** Stream results for the developed process.

| Component | Water | Methanol | EO | ROUT | MEG | DEG | TEG |
|---|---|---|---|---|---|---|---|
| Pressure (kPa) | 675.7 | 101.35 | 101.34 | 3495.6 | 101.35 | 101.35 | 103.4 |
| Temperature (°C) | 162.2 | 63.9 | 20.6 | 204.4 | 136.7 | 123.3 | 224.4 |
| Flow rate(kg/h) | 29,917.6 | 103,211.3 | 22,809.4 | 315,195.4 | 30,873.1 | 1020.2 | 19.5 |
| *Composition (mass%)* | | | | | | | |
| Hydrogen peroxide | 0 | 0 | 0 | 0 | 0 | 0 | 0 |
| Methanol | 0.01 | 99.51 | 0.06 | 0 | 0 | 0 | 0 |
| Ethylene | 0 | 0.48 | 0 | 0 | 0 | 0 | 0 |
| Ethylene oxide | 0 | 0 | 99.93 | 0.22 | 0 | 0 | 0 |
| Oxygen | 0 | 0 | 0 | 3.80 | 0 | 0 | 0 |
| Water | 99.98 | 0 | 0 | 85.41 | 0.01 | 0 | 0 |
| MEG | 0 | 0 | 0 | 10.23 | 99.99 | 0.04 | 0 |
| DEG | 0 | 0 | 0 | 0.33 | 0 | 99.95 | 3.22 |
| TEG | 0 | 0 | 0 | 0.006 | 0 | 0 | 96.77 |
| **Total** | **100** | **100** | **100** | **100** | **100** | **100** | **100** |

The heat integration results are tabulated in Table 12. When individual heat integration is considered within two sub-plants (ethylene to EO and EO to EG) but there is no integration between these sub-plants, net savings of 36.1% of hot utilities and 31.4% of cold utility are achieved when compared to the base case in which there is no heat integration. When heat integration between the two sub-plants is included, net savings of 40.0% in hot utilities and 34.8% in cold utilities are achieved. This simulation demonstrates that a substantial amount of utility cost can be saved via heat integration. The optimized total utility cost for this model is USD 4,830.36 USD per hour, which is summarized in Table 13. The required hot utility is provided by HP steam for 274.64 MW, and the cold utility is 343.13 MW, which is provided by cooling water for 320.91 MW and refrigerant for 22.22 MW. The final HEN design, which satisfies all thermodynamic matching and heat load requirements presented in Table 3, is shown in Figure 5. If the hot and cold utilities are independently supplied for the process without the consideration of heat integration, the total amount of hot and cold utility duties will be 457.93 MW and 526.05 MW instead of 274.64 MW and 342.89 MW, respectively. Figure 5 shows only one of the possible optimal designs, although other solutions are also possible.

**Table 12.** Comparison of HEN performance for subprocesses.

| Classification of Heat Integration (HI) | Hot Utility (MW) | Cold Utility (MW) | Savings (%) | |
|---|---|---|---|---|
| | | | Hot Utility | Cold Utility |
| Base case: no HI | 457.93 | 526.05 | – | – |
| HI within two sub-plants | 292.60 | 360.77 | 36.1 | 31.4 |
| HI over entire plant | 274.64 | 342.89 | 40.0 | 34.8 |

**Table 13.** Optimized utility cost.

| Utilities | Optimized Utility (MW) | Optimized Cost (USD/h) |
|---|---|---|
| Cooling Water | 320.91 | 244.92 |
| Refrigerant | 22.22 | 381.52 |
| HP Steam | 274.64 | 4203.92 |
| **Total** | | **4830.36** |

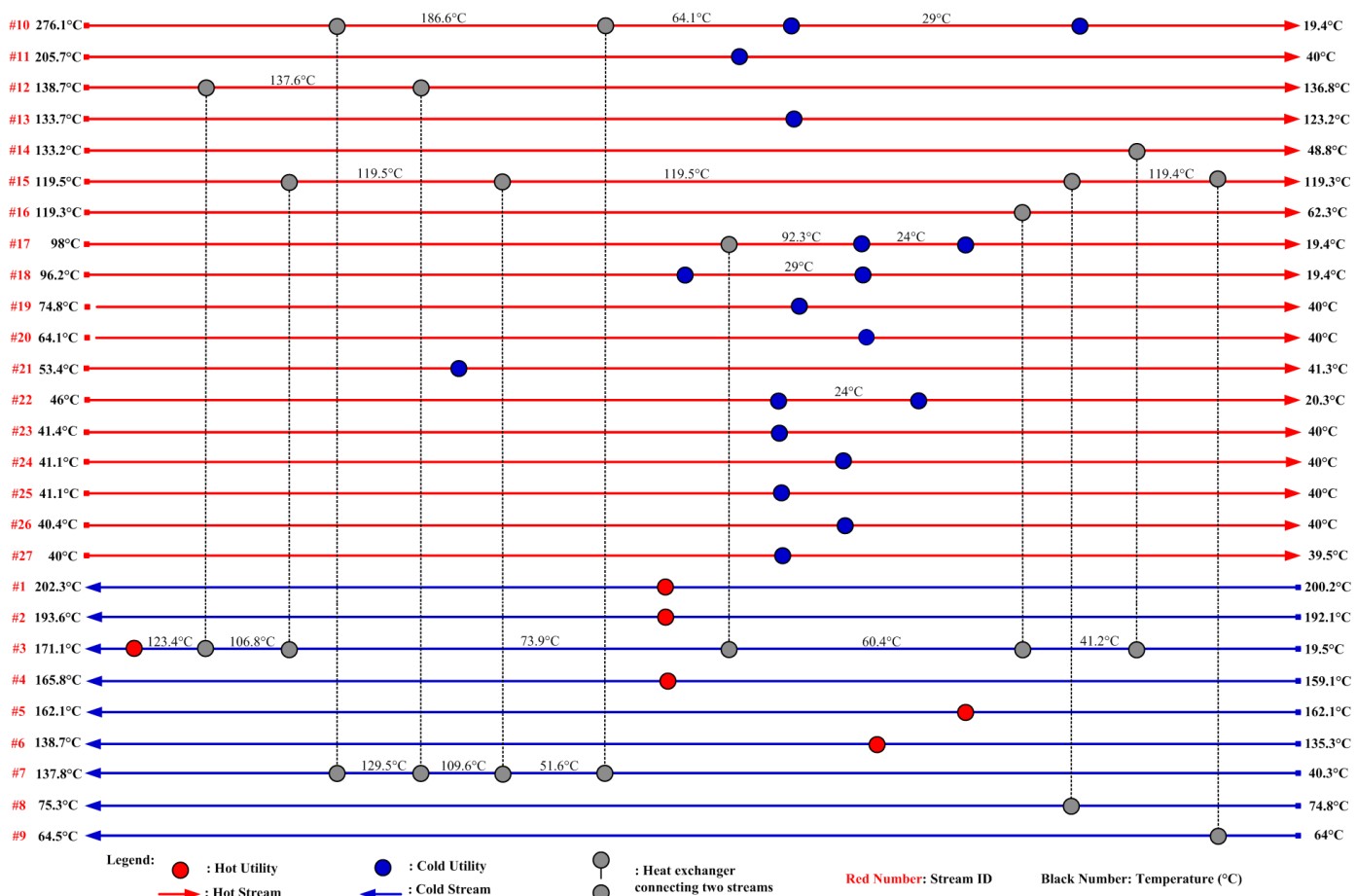

**Figure 5.** HEN grid diagram for the EG process.

We conducted an economic analysis of the conventional process and the new process for ethylene oxide and MEG production. It was assumed that the plant produces 200,000 t/y of MEG, and the results of total capital investment for the EO process and the MEG process are shown separately in Tables 14 and 15, respectively. The catalyst life was assumed to be 1 year, with a leach rate of 0.05 kg MTO/h. Based on the raw material, utility costs (Tables 8 and 13), and fixed capital investment, the total initial investment required for the development of the newly developed ethylene-to-EO process is USD 90,611,300. In comparison, the conventional epoxidation process cost is USD 107,441,468, and the process developed by the Center for Environmental Beneficial Catalysis (CEBC) at the University of Kansas, requires an initial investment of USD 109,238,636 [37]. Similarly, the total initial investment for this modified EO-to-MEG production process is USD 16,729,700, which is significantly less than the total initial investment of USD 20,598,400 required for the conventional EO-to-MEG production process [38]. These simulation results demonstrate that process integration techniques can improve the economics of such a plant while maintaining the desired product specifications.

**Table 14.** Comparison of total capital investment for the developed EO process.

| Description | Conventional Process Cost (USD)[37] | CEBC Process Cost (USD)[37] | New EO Process Cost (USD)[14] |
|---|---|---|---|
| Reactors | 11,289,812 | 6,045,242 | 2,613,000 |
| Columns | 7,309,166 | 5,250,531 | 58,116,000 |
| Compressors | 8,116,280 | 3,175,809 | 2,429,000 |
| Heat exchangers | 7,658,600 | 15,621,345 | 1,555,000 |
| Pumps | 1,101,300 | 1,848,500 | 988,000 |
| Vessels and tanks | 2,081,000 | 5,968,669 | 539,000 |
| Auxiliaries | 69,885,310 | 71,328,540 | 24,371,300 |
| Total fixed capital investment | 107,441,468 | 109,238,636 | 90,611,300 |

**Table 15.** Comparison of total capital investment required for the modified MEG process vs. the conventional process.

| Description | Conventional Process Cost (USD)[38] | Modified Process Cost (USD) |
|---|---|---|
| Reactors | 74,400 | 73,500 |
| Columns | 4,141,600 | 4,101,500 |
| Heat exchangers | 1,859,300 | 1,865,300 |
| Pumps | 331,200 | 331,200 |
| Vessels and tanks | 890,500 | 870,500 |
| Auxiliaries | 13,301,400 | 9,487,700 |
| Total fixed capital investment | 20,598,400 | 16,729,700 |

## 4. Conclusions

In this paper, a new MTO-based EG production process was analyzed via simulation. This new process is potentially safer and more profitable than conventional industrial practices with respect to gas-phase epoxidation. The modified EG process considered in this research represents the following innovations: (i) A multitube reactor was used for the purpose of energy saving and cost cutting, and (ii) an ethylene-to-EO plant is integrated with an EO-to-EG plant to reduce utility costs and recover high-value products. The resultant plant layout and operating conditions have the potential to reduce the production costs of EG. The integrated EO/EG plant produces ethylene glycols with the same purity and product recovery as the conventional process. A comparative economic assessment based on a 200,000 t/y plant indicates that process integration techniques can reduce costs significantly.

**Author Contributions:** Conceptualization, S.P. and Q.X.; methodology, M.E.H. and N.T.; software, M.E.H. and N.T.; validation, M.E.H., K.D.P.N. and N.T.; formal analysis, S.P., M.E.H. and N.T.; investigation, S.P., M.E.H., N.T.; resources, S.P., data curation, M.E.H. and S.P., writing-original draft preparation, S.P., N.T., K.D.P.N., and Q.X.; writing-review and editing, S.P. and M.E.H., visualization, M.E.H., N.T., supervision, S.P., project administration, S.P.; funding acquisition, S.P. All authors have read and agreed to the published version of the manuscript.

**Funding:** This work is supported in part by funding from the National Institutes of Standards and Technology (Award 70NANB15H047) under the Manufacturing Extension Partnership Program.

**Institutional Review Board Statement:** Not applicable.

**Conflicts of Interest:** The authors declare no conflict of interest.

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
