# Peer review of "Plant-Wide Modeling and Economic Analysis of Monoethylene Glycol Production"

_processes, doi:10.3390/pr10091755_

Round 1

Reviewer 1 Report

In article "Plant-wide Modeling and Economic Analysis for the Production of Monoethylene Glycol” authors present rather full modeling scheme, calculated in Aspen. Authors show high necessity of MEG. On the other hand, I’ve seen no new methods, ideas etc. Thus, scientific soundness is rather low in my opinion.

Being objective, calculating methods rarely has any scientific soundness. So, I recommend this article for publication.

Author Response

We would like to thank the reviewer for recommending publication. Minor spelling errors have been corrected. 

Reviewer 2 Report

Τhe keywords presented in the manuscript are only few and not easy to be searched in my opinion. I believe they should be enriched with more plain words to be easily detectable. In the second paragraph of introduction, the phrase "from of 3.6%"
needs to be improved/corrected. Similar grammatical/syntactical and type errors should be corrected in the whole text. The topic should be more thoroughly approached in the introduction chapter and the state-of-the-art should be described in detail. If possible the number of references should be increased. The authors are highly recommended to incorporate as well in the introduction the recent work of DOI: 10.3390/su132212810 to support the theoretical part. It is not very clear where the materials-methods chapter begins, where it ends and where the results chapter begins. The production process has not been properly/thoroughly described and it could be improved to be easily comprehensive to the reader. The recommended format of the journal has not been applied in the test, especially in the references list.

Author Response

We would like to thank the reviewer for reading the manuscript and giving valuable feedback, which has substantially improved the quality of the manuscript. The points raised by the reviewer have been fully addressed in the revised manuscript as described below:

Comment 1:

Keywords presented in the manuscript are only few and not easy to be searched in my opinion. I believe they should be enriched with more plain words to be easily detectable.

Response:

Thank you for pointing out this issue. We have added two additional keywords: “Process Modeling, Process Integration,” in our revised manuscript.

Comment 2:

In the second paragraph of introduction, the phrase "from of 3.6%" needs to be improved/corrected. Similar grammatical/syntactical and type errors should be corrected in the whole text.

Response:

Thanks for raising this issue. We have checked our manuscript thoroughly for grammatical errors and corrected in the revised manuscript.

Comment 3:

The topic should be more thoroughly approached in the introduction chapter and the state-of-the-art should be described in detail. If possible the number of references should be increased. The authors are highly recommended to incorporate as well in the introduction the recent work of DOI: 10.3390/su132212810 to support the theoretical part.

Response:

As suggested by the reviewer, we have added the following text in the introduction part and added several more reference.

“A conventional approach of industrial production of EO from ethylene is epoxidation reaction explained by Rebsdat and Mayer, 2012 [2]. A useful summary of this ethylene oxidation reaction system can be found at Nawaz et al., 2016 [8], and most studies have used the Langmuir-Hinshelwood-Hougen-Watson (LHHW) type kinetics. The LHHW approach assumes that all active sites are energetically uniform, and, upon adsorption, adsorbed species do not interact with species already adsorbed. Active sites have similar kinetic and thermodynamic characteristics, and the entropy and enthalpy of adsorption are constant and not functions of the adsorbed amount. The species adsorption restricts itself to only monolayer coverage and the rate of adsorption is proportional to the concentration of the active sites not occupied (empty) and the partial pressure of the component in the gas phase [9,10]. In 1990s, the Westerterp group published a series of papers where extensive experiments and kinetic model development for EO are presented. Their study focused on kinetics of ethylene in the presence of excess air on an unpromoted silver catalyst supported on alumina in the absence of chlorinated hydrocarbon moderators [9-12].”

However, we did not cite the reference suggested by reviewer, which is on “Anaerobic Digestion of Lignocellulosic Waste Materials” and is not related to our research.

Comment 4:

It is not very clear where the materials-methods chapter begins, where it ends and where the results chapter begins. The production process has not been properly/thoroughly described and it could be improved to be easily comprehensive to the reader. The recommended format of the journal has not been applied in the test, especially in the references list.

Response:

In the revised manuscript, we have improved the process description part, provided separate feed specifications and product stream results, and corrected some reference which were not as per journals format in our revised manuscript. We have organized our paper sequentially as, Introduction, Process Description and Simulation, Simulation Results and Discussion and Conclusion. In the Introduction section, we have written about ethylene oxide and ethylene glycol’s global market trend, hazards associated in the process, cost reduction opportunity and research scope. In the Process Description section, we discussed about process modeling in detail and provided a description of the process, feed specifications, reaction kinetics, heat integration model, equipment sizing and cost analysis of the process. In the Simulation Results and Discussion section, we have discussed about reactor composition profiles, column composition profiles, heat exchange network results and the final NPV results.

Reviewer 3 Report

The paper presents the use of some Aspentech modelling tools on the EO/EG process, claiming novelty on the energy integration of the EO and EG plants, and the use of a multi-tubular reactor in the glycol plant.

This is a bad start: EO and EG plants are commonly integrated (just see Shell processes) due to the favourable energy balance and the reduction in transportation costs (and risks) of EO, which major end use is EG production. The reactor proposal is ridiculous: the authors compare the multitube version with a reactor which diameter is the same of one of the tubes, resulting in an unnecessary long equipment. Just use the same join section to determine the diameter and you got the same length and production at a reduced cost.

In general, this is an unprofessional academic exercise. The process flowsheet is a bad quality print of the process simulation window, but the material and energy balance table with the information of the streams is absent. In table 3, just some component flowrates are shown, but ROUT is unreferenced. Also, sub-flowsheets are simply declared.

It is not clear how the UNIQUAC/PR thermodynamic model is used for V/L equilibrium: Ethylene Is over its critical temperature in the EO plant, so you cannot consider UNIQUAQ.

Part of the text regarding the EO reactor seems to prey on the work of Ghanta et al. The pinch analysis is a routine HEN design (that does not include the first evaporator!) made with Aspen Energy Analyzer. But the optimization of the energy integration of the EO and EG plants is much more than this. It must, at least, consider EG column design.

Formally, there are font changes (Palatino and Times) in the middle of the text. Profile plots I figures 2 to 5 have low information density.

I see more problems, but I have cited more than enough to reject the manuscript.

Author Response

General Comments:

The paper presents the use of some Aspentech modelling tools on the EO/EG process, claiming novelty on the energy integration of the EO and EG plants, and the use of a multi-tubular reactor in the glycol plant.

This is a bad start: EO and EG plants are commonly integrated (just see Shell processes) due to the favourable energy balance and the reduction in transportation costs (and risks) of EO, which major end use is EG production. The reactor proposal is ridiculous: the authors compare the multitube version with a reactor which diameter is the same of one of the tubes, resulting in an unnecessary long equipment. Just use the same join section to determine the diameter and you got the same length and production at a reduced cost.

Response:

We would like to thank the reviewer for reading the manuscript and giving valuable feedback, which has substantially improved the quality of the manuscript. While it is true that EO and EG plants are integrated, we are not aware of any studies in the published literature about quantifying the economic benefit of doing this. Our conversations with our industry partners indicate that this was generally being done for reasons of safety (i.e. the difficulty in transporting EO). The focus of our study to show the economic benefits of an integrated plant. For clarity, we have also removed the conventional reactor part and corrected all necessary parameters in the revised manuscript.

Comment 1:

In general, this is an unprofessional academic exercise. The process flowsheet is a bad quality print of the process simulation window, but the material and energy balance table with the information of the streams is absent. In table 3, just some component flowrates are shown, but ROUT is unreferenced. Also, sub-flowsheets are simply declared.

Response:  

As suggested by the reviewer, we have improved the process flowsheet in our revised manuscript. In Table 3, we only provided the feed and product stream data. We did not provide all the stream data to avoid providing unnecessary details that would take the focus away from the main point of the manuscript. We did provide separate tables for feed specification (Table 1) and product stream results (Table 11). The ROUT stream is now mentioned in the PFD in our revised manuscript. We added a description for sub-flowsheet REGEN-H2O2 HIERARCHY and PUMP HIERARCHY in the process description part as:

A series of four pumps and four heaters are used to raise the stream pressure and temperature to the required feed condition in the solvent recycle stream. These are represented in the PUMP sub-flowsheet block. A conversion reactor is utilized to convert the generated oxygen to hydrogen peroxide that is recycled that back to first reactor. A heater block is used to raise the stream pressure and temperature to operation conditions. This is represented by the REG-H2O2sub-flowsheet block.”

Comment 2:

It is not clear how the UNIQUAC/PR thermodynamic model is used for V/L equilibrium: Ethylene Is over its critical temperature in the EO plant, so you cannot consider UNIQUAQ.

Response

The reviewer is correct about the inapplicability of UNIQUAQ when ethylene is over the critical temperature. In the revised manuscript, we have used the NRTL thermodynamic package for ethylene oxide production and CPA thermodynamic package for glycol production. There were minor changes in the results, and we have updated these changes in the revised manuscript.

Comment 3:

Part of the text regarding the EO reactor seems to prey on the work of Ghanta et al. The pinch analysis is a routine HEN design (that does not include the first evaporator!) made with Aspen Energy Analyzer. But the optimization of the energy integration of the EO and EG plants is much more than this. It must, at least, consider EG column design.

Response:

In the heat exchange network, we did not consider three evaporators (EVA-1, EVA-2, & EVA-3) as this part is already heat integrated. We also provided the column sizing parameters in the Table 4.

Comment 4:

Formally, there are font changes (Palatino and Times) in the middle of the text. Profile plots I figures 2 to 5 have low information density. I see more problems, but I have cited more than enough to reject the manuscript.

Response:

We have checked our manuscript thoroughly and found no font changes occur. This might have happened during conversion from word to pdf format in the journal.

Reviewer 4 Report

Comments and suggestions for authors are presented in detail in the attached file.

The first general remark is related to the use of imperial units which should be changed to SI units and the second one is to separate the economic part into new subtitle and arrange it according to CAPEX and OPEX methodology.

Author Response

General Comments: Interesting idea for upgrading process for the production of MEG and integration with EO tominimize investment, optimize utilities and energy consumption.

Response:

We would like to thank the reviewer for reading the manuscript and giving valuable feedback, which has substantially improved the quality of the manuscript. The points raised by the reviewer have been fully addressed in the revised manuscript as described below.

Comment 1:

Generally, units need to be transformed into SI units

Response:

We chose to use “British Units” in our simulation because these units are commonly used in the U.S. industry.

Comment 2:

Some abbreviations are not explained

Response:

These issues have been corrected in our revised manuscript.

Comment 3:

Figure 1. is of bad quality and should be changed; the same is for Fig. 4, 5 etc.

Response:

As suggested by the reviewer, we have improved and corrected Figure 1, 4 & 5.

Comment 4:

In the Fig.1 Q1, Q2 and Q3 are not explained despite some explanations in the text

Response:

We have put a legend in the PFD to indicate that the red dotted lines represented by Q1, Q2 and Q3 are heat streams.

Comment 5:

Langmuir–Hinshelwood–Hougen–Watson (LHHW) type of reaction should be described somewhere

Response:

We have added the following text in to the Introduction part, which provides a brief description of LHHW reactions as well as appropriate references that give the mathematical form of the reaction:

“A conventional approach of industrial production of EO from ethylene is epoxidation reaction explained by Rebsdat and Mayer, 2012 [2]. A useful summary of this ethylene oxidation reaction system can be found at Nawaz et al., 2016 [8], and most studies have used the Langmuir-Hinshelwood-Hougen-Watson (LHHW) type kinetics. The LHHW approach assumes that all active sites are energetically uniform, and, upon adsorption, adsorbed species do not interact with species already adsorbed. Active sites have similar kinetic and thermodynamic characteristics, and the entropy and enthalpy of adsorption are constant and not functions of the adsorbed amount. The species adsorption restricts itself to only monolayer coverage and the rate of adsorption is proportional to the concentration of the active sites not occupied (empty) and the partial pressure of the component in the gas phase [9,10]. In 1990s, Westerterp group published a series of papers where extensive experiments and kinetic model development for EO are presented. Their study focused on kinetics of ethylene in the presence of excess air on an unpromoted silver catalyst supported on alumina in the absence of chlorinated hydrocarbon moderators [9-12].”

Comment 6:

Some inconsistencies were noticed using the thousand (comma) separator, which should be used elsewhere

Response:

These have been corrected in the revised manuscript. Thank you for raising this issue.

Comment 7:

Table 13. R1 reactor is not assigned

Response:

This has been corrected in the revised manuscript.

Comment 8:

The economic part is not presented transparently enough; use CAPEX and OPEX principles to clearly present economic features. Probably new subtitle can be used for economic analysis

Response:

As suggested by the reviewer, we have separated out the capital cost (CAPEX) and the operating cost (OPEX) and indicated clearly which cost we are using for our comparative analysis.

Comment 9:

Costs are not simply costs; they should be differentiated between Expenditures (CAPEX) and Expenses (OPEX)

Response:

As mentioned above, we have separated out the capital cost (CAPEX) and the operating cost (OPEX) and indicated clearly which cost we are using for our comparative analysis as indiacted in Table 9 and Table 14.

Comment 10:

Table 17. Is it not explicit which economic performances are presented (OPEX ?)

Response:

As suggested by the reviewer, in the revised version we have explicitly mentioned that we are comparing the capital cost in Table 15 (Table 17 in the previous version of the manuscript). Furthermore, we have modified this table so that it is similar to Table 14 and provides more details of the individual equipment costs.

Round 2

Reviewer 2 Report

The most of the proposed changes in the revised verion of manuscript towards the improvement of their work have ibeen mplemented by authors.  The manuscript is quite well-prepared and organized enough in the present form.They have enrich the keywords and they have add relevant references to the introduction.Further improvement and a general check should be implemented for syntactical and type errors in the whole text.

I remain at your disposal for any clarification.

Author Response

We would like to thank the reviewer for positive comments. We have gone over all the sections and improved the presentation by correcting style and grammatical errors. 

Reviewer 4 Report

The manuscript is correctly improved acc. to my observations.

The decision about SI or imperial units is up to Editor!

There are some minor issues to be corrected:

- page1, not capital expenses (CAPEX) but: capital expenditures (expenses are only at OPEX, find the difference on google!)

- p. 2, NRTL is not explained

- table 6  Design temperature oF not only F

OK

Author Response

We would like to thank the reviewer for positive comments. We have made the suggested changes in page 1 (changed "expense" to "expenditure"), page 2 (defined the full form of NRTL) and Table 6 (changed F to oF).